# Biofungicides Based on Plant Extracts: On the Road to Organic Farming

**DOI:** 10.3390/ijms25136879

**Published:** 2024-06-22

**Authors:** Antonio de Jesús Cenobio-Galindo, Alma Delia Hernández-Fuentes, Uriel González-Lemus, Ana Karen Zaldívar-Ortega, Lucio González-Montiel, Alfredo Madariaga-Navarrete, Iridiam Hernández-Soto

**Affiliations:** 1Instituto de Ciencias Agropecuarias, Universidad Autónoma del Estado de Hidalgo, Av. Universidad Km 1 Rancho Universitario, Tulancingo 43600, Hidalgo, Mexico; antonio_cenobio@uaeh.edu.mx (A.d.J.C.-G.); almah@uaeh.edu.mx (A.D.H.-F.); uriel_gonzalez@uaeh.edu.mx (U.G.-L.); ana_saldivar@uaeh.edu.mx (A.K.Z.-O.); alfredo_madariaga@uaeh.edu.mx (A.M.-N.); 2Instituto de Tecnología de los Alimentos, Universidad de la Cañada, Teotitlán de Flores Magón 68540, Oaxaca, Mexico; luciogonzalez@unca.edu.mx

**Keywords:** phytopathogenic fungi, secondary metabolites, agricultural sustainability, agroecology, phytosanitary products

## Abstract

Phytopathogenic fungi are responsible for diseases in commercially important crops and cause major supply problems in the global food chain. Plants were able to protect themselves from disease before humans played an active role in protecting plants. They are known to synthesize a variety of secondary metabolites (SMs), such as terpenes, alkaloids, and phenolic compounds, which can be extracted using conventional and unconventional techniques to formulate biofungicides; plant extracts have antifungal activity and various mechanisms of action against these organisms. In addition, they are considered non-phytotoxic and potentially effective in disease control. They are a sustainable and economically viable alternative for use in agriculture, which is why biofungicides are increasingly recognized as an attractive option to solve the problems caused by synthetic fungicides. Currently, organic farming continues to grow, highlighting the importance of developing environmentally friendly alternatives for crop production. This review provides a compilation of the literature on biosynthesis, mechanisms of action of secondary metabolites against phytopathogens, extraction techniques and formulation of biofungicides, biological activity of plant extracts on phytopathogenic fungi, regulation, advantages, disadvantages and an overview of the current use of biofungicides in agriculture.

## 1. Introduction

Agricultural crops are constantly exposed to and/or threatened by diseases caused by phytopathogenic fungi that affect their growth and quality. There are reports stating that diseases account for between 20 and 40% of crop losses [1]. The ultimate impact of the losses takes the form of insufficient food production and chronic hunger. The severity of this problem is rapidly being exacerbated by increasing urbanization, climate change, and the emergence of phytopathogens resistant to commercial fungicides [2]. According to a recent assessment of the projections of the United Nations Department of Economic and Social Affairs report, the current world population will increase from 7.3 billion to 8.5 billion people by 2030; it will reach more than 9.5 billion in 2050 and more than 11 billion in 2100 [3]. Therefore, agricultural production must be improved to meet the demands of the rapidly growing world population, as agriculture is the most important economic factor for a healthy and sustainable society [4]. However, to date, synthetic fungicides are mainly used to control these diseases, which cause damage to health and the environment [5]. In addition to the total consumption, about 10% of fungicides reach the target organism, while 90% remain in the environment [6]. In this context, organic agriculture is the most sustainable response to current crises of all kinds, as it can better anticipate and prepare for crises and create long-term equity and resilience in food systems [7].

Plants have been able to protect themselves from diseases before humans played an active role in protecting them. They are known to synthesize a variety of secondary metabolites (SMs) with a specific biological activity [8]. These are categorized into different chemical groups according to their biosynthetic pathways: terpenes (including volatile compounds, sterols, and carotenoids), polysaccharides, phenolic compounds, phytoalexins (sulfur-containing compounds), alkaloids (nitrogen-containing compounds), flavonoids, and hydrocarbons [9]. Not all secondary metabolites are found in all plant groups; they are synthesized in small amounts and not in general forms, and their production is often restricted to a particular plant genus, family, or even some species [10]. SMs are extracted from plants to produce natural products using conventional and unconventional techniques and specific solvents, depending on the group of phytochemicals of interest [11]. Due to their natural origin, they are referred to as environmentally friendly agents. They have a limited persistence in the field and a shorter useful life, no residual hazards, and minimize the pollution of soil, water, and the atmosphere [12,13]. There are even reports suggesting that they have an ecological function that is reflected in a reduction in respiration by creating a specific microclimate that protects plants from excessive transpiration, reflection, and refraction of light, which allows the plant to adapt its immediate environment to create less favorable conditions for pathogens and thus improve its survivability, and finally, they promote a reduction in health problems in farmers, such as chronic degenerative diseases of the skin and respiratory tract associated with the use of synthetic pesticides [14,15]. These biofungicides have various mechanisms of action, including the inhibition of germ tube elongation, delaying sporulation, DNA damage, inhibition of protein synthesis, damaging the structures of hyphae and mycelia, inhibition of the production of toxic substances from mycotoxin-producing fungi, and many more [16]. Currently, there is a global trend towards the consumption of food produced with organic products. The discovery of dangerous residues of chemical fungicides in food and increased consumer awareness of food safety have led to the banning of certain fungicides in agricultural production and the increasing popularity of plant-based fungicides in agriculture [17]. Therefore, biofungicides are proposed as safe for use on crops for human consumption and there is currently a lucrative market among consumers who are willing to pay more for organically produced food [18]. Therefore, it was considered relevant to conduct a literature review with the purpose of collecting data on biosynthesis; the mechanisms of action of secondary metabolites against phytopathogens; extraction techniques and formulation of biofungicides; the biological activity of plant extracts on phytopathogenic fungi; and the regulation, advantages, disadvantages, and an overview of the current use of biofungicides in agriculture.

## 2. Biosynthesis of Secondary Metabolites in Plants and Their Mechanisms of Action against Phytopathogenic Fungi

Plants naturally produce a variety of products of different chemical natures that are used for plant growth and development. Primary metabolites provide the necessary substances for processes such as photosynthesis, translocation, and respiration. The products from primary metabolism that are not directly involved in growth and development are called secondary metabolites [19]. There are three main types of secondary metabolites according to their biosynthesis: (a) terpenes and terpenoids, (b) alkaloids, and (c) phenolic compounds, and there are four main pathways for the synthesis of these metabolites: (1) the shikimic acid pathway, (2) the malonic acid pathway, (3) the mevalonic acid pathway, and (4) the non-mevalonate (MEP) pathway (Figure 1) [20].

## 3. Terpenes

Terpenes are synthesized via the mevalonic acid pathway and the 2-C-methyl-d-erythritol 4-phosphate (MEP) pathway [21]. All terpenoids are derived from the two 5-carbon building blocks isopentenyl diphosphate (IPP) and its enzymatically convertible isomer dimethylallyl diphosphate (DMAPP). These C5 compounds are generated via both the 2-C-methyl-d-erythritol-4-phosphate (MEP) pathway, which occurs in the plastids, and the mevalonate (MVA) pathway, which is distributed between the cytoplasm, the endoplasmic reticulum, and the peroxisomes [22].

## 4. Alkaloids

Alkaloids are a class of naturally occurring nitrogen-containing organic compounds [23]. Alkaloids have been reported to be present in most dicotyledonous plants, such as plants from the families Ranunculaceae, Papaveraceae, Solanaceae, Menispermaceae, Berberidaceae, and Leguminosae [24]. Alkaloids are produced by aromatic amino acids (they originate from the shikimic acid pathway) and by aliphatic amino acids (they originate from the tricarboxylic acid cycle) [25].

## 5. Phenolic Compounds

Phenolic compounds are carbon-based phytochemicals found in plants [26]. Chemically, phenolic compounds are a very diverse group, ranging from simple molecules such as phenolic acids to complex polymers such as condensed tannins and lignin. Flavonoid pigments are also found within phenolic compounds [27]. The biosynthesis of phenolic compounds occurs via the shikimic acid pathway [28].

The mechanisms of action in phytopathogenic fungi are variable (Figure 2); it is assumed that terpenes alter the cell membranes of fungi. This membrane, which consists of a lipid bilayer containing phospholipids and ergosterol, can be destabilized by terpenes due to their lipophilic nature. The incorporation of terpenes into the membrane alters its fluidity and permeability, affecting ion gradients and other important functions of the cell membrane. In addition, terpenes can destroy mitochondria, the organelles responsible for energy production in the cell through cellular respiration and oxidative phosphorylation. By destabilizing the inner and outer mitochondrial membranes, terpenes prevent the proper production of ATP and can release pro-apoptotic compounds that induce cell death [29]. Another crucial effect of terpenes is the inhibition of electron transport in the mitochondrial electron transport chain. This process, which takes place in the inner membrane of the mitochondria, is essential for the creation of a proton gradient that drives ATP synthesis. Terpenes interfere with the protein complexes involved in this transport and stop the creation of the proton gradient and thus the production of ATP. Finally, terpenes also inhibit ATPase, an enzyme that synthesizes ATP from ADP and inorganic phosphate, with the help of the proton gradient generated by the electron transport chain [30,31].

It is speculated that phenolic compounds affect the cellular and mitochondrial membranes of fungi and cause their depolarization. This depolarization impairs the ion gradients that are important for energy production and cellular homeostasis, leading to dysfunction and eventual cell death. In addition, phenolic compounds inhibit key enzymes such as chitinases [32]. The inhibition of chitinases prevents the proper maintenance and remodeling of the cell wall, thus weakening the structure of the fungus. Another important mechanism of action is the ability of phenolic compounds to modulate gene expression in fungi. This modulation affects vital processes of the pathogen, such as the synthesis of important proteins and the regulation of the cell cycle, significantly altering the growth, development, and reproduction of the fungus [33].

Alkaloids exert their antimicrobial effect via various biochemical mechanisms. Intercalation with microbial DNA impairs gene replication and transcription, which impairs the ability of the microorganism to reproduce and synthesize important proteins [15] (Figure 2). In addition, alkaloids can form ion channels in the microbial membrane, disrupting ion gradients and cellular homeostasis, leading to cellular dysfunction and death. These compounds also cause competitive inhibition by attaching to microbial proteins, impairing their ability to interact with host receptor polysaccharides, thus preventing colonization and infection [34].

Kang et al. [35] investigated the antifungal activity of baicalein (a type of flavonoid) against *Candida krusei*. Baicalein showed strong antifungal activity against this pathogen in vitro. In this study, baicalein was reported to depolarize the mitochondrial membrane potential, which led to the conclusion that the antifungal activity of baicalein in *C. krusei* occurs through mitochondrial disruption. Tian et al. [36] investigated the antifungal activity of dill (*Anethum graveolens* L.) essential oil against *Aspergillus flavus*. It was reported that the essential oil causes morphological changes in the cells of *A. flavus* and a reduction in the amount of ergosterol (a lipid component of the membrane). On the other hand, changes in mitochondrial membrane potential, acidification of the external environment, and changes in mitochondrial ATpase and dehydrogenase activities were found. Wu et al. [37] investigated the effect of plagiokine E (a macrocyclic bis(bibenzyl) antifungal agent) isolated from Marchantia polymorpha L (PLE) on chitin synthesis in the cell wall of *Candida albicans*. According to the authors’ report, the antifungal activity of PLE is due to its inhibitory effect on the synthesis of chitin in the cell wall of *Candida albicans*. Bagiu et al. [38] analyzed the activity of *Allium ursinum* L. extracts against *Candida* spp. (*C. albicans*, *C. fameta*, *C. glabrata*, and *C. krusei*). The activity was attributed to allicin sulfoxide and S-methylcysteine as well as mixtures of volatile substances produced by these two compounds. It is important to note that in sulfoxide, the sulfur atom is linked to an oxygen atom and to two different organic groups (R and R′) via a double bond (S=O). This arrangement can lead to a chiral center in sulfur, with two possible stereoisomeric configurations (enantiomers). This chirality implies that sulfoxide enantiomers can have different biological activities. In the case of allicin sulfoxide, stereochemistry plays a crucial role due to the possible chirality of the sulfur atom, which can adopt R or (S) configurations. When it adopts the (S) configuration, it reportedly uncouples the protons and inhibits the translocation of protons to a membrane vesicle and subsequently disrupts ADP phosphorylation [39,40,41], also interfering with enzyme proteins that are integrated into or associated with the membrane, stopping their production or activity. Sulfur compounds also inhibit the synthesis of DNA, RNA, proteins, and polysaccharides in fungal and bacterial cells [42] (Figure 2).

Some mechanisms of biofungicides are similar to those of chemical fungicides; for example, benzimidazoles have a chemical structure similar to the bases of the nucleic acids of fungi. They can replace the bases of nucleotides and prevent the polymerization of nucleotides into nucleic acids and thus influence nucleotide synthesis in fungi [16]. Acylalanines inhibit ribosomal RNA synthesis; carbamate vanilamides inhibit cell wall biosynthesis; strobilurins, oxazolidinones, and imidazolones inhibit mitochondrial respiration; and toluamides inhibit cell division, to name but a few [43].

## 6. Methods for Extraction of Secondary Metabolites and Formulation of Biofungicides

Plants represent a valuable source of chemical compounds that are used to develop new products [44]. Given the large differences between chemical compounds and the large number of plant species, it is necessary to develop a standardized and integrated approach to extract such compounds. Most of these techniques are based on the extraction power of the different solvents used and the application of heat and/or mixing [45]. To obtain these compounds, conventional and unconventional techniques are used (Table 1) [46]. The extraction efficiency of each method depends mainly on the choice of solvent. The polarity of the target compound is the most important factor in the choice of solvent. The molecular affinity between solvent and solute, mass transfer, environmental safety, human toxicity, and financial viability should also be considered when selecting a solvent for the extraction of bioactive compounds. Some examples of bioactive compounds extracted with different solvents can be found in Figure 3 [47,48,49,50,51].

## 7. Conventional Techniques

### 7.1. Soxhlet Extraction

Soxhlet extraction was originally developed for the extraction of lipids. Currently, its use is not limited to lipid compounds, but it is also widely used for the extraction of active ingredients from various natural sources [52]. Soxhlet extraction consists of the following steps: (1) A small amount of the dry sample is placed in a thimble (a porous support made of filter paper or cellulose). (2) The thimble is then placed in a distillation flask containing the particular solvent that depends on the phytochemicals to be extracted. (3) After reaching an overflow level, the solution is drawn from the thimble into a siphon, which drains the solution back into the distillation flask. (4) This solution carries extracted solutes into the main liquid. The solutes remain in the distillation flask and the solvent returns to the fixed bed of the unit. (5) The process is repeated until extraction is complete [53].

### 7.2. Maceration

In solid–liquid extraction, often referred to as maceration, the solvent has a major influence on selectivity. Its polarity has a direct influence on the extracted solutes, which is related to the chemical structure of the compounds. Modeling the interactions between the compound and solvent using various scales of polarity or interaction is a major challenge to favor the choice of the appropriate extraction liquid [54]. Maceration consists of three main steps: (1) the plant sample is crushed into small particles to increase the surface area with the solvent; (2) the subsequent addition of a suitable solvent in a closed container, preferably amber in color to avoid the photo-oxidation of the phytochemicals; and (3) the liquid is filtered to recover a large amount of entrapped solutions [55].

### 7.3. Hydrodistillation

Hydrodistillation is a traditional method for extracting active ingredients and essential oils from plant samples. This process is recommended for the extraction of volatile active ingredients, but since a high extraction temperature is required, it cannot be used for thermolabile compounds. In addition, hydrodistillation is a technique that does not require organic solvents and can be applied to both dry and wet plant samples [56]. Hydrodistillation involves three main physicochemical processes: hydrodiffusion, hydrolysis, and heat decomposition. Firstly, the plant materials are packed into a distillation compartment; secondly, water is added in a sufficient quantity and then brought to a boil. Alternatively, direct steam is injected into the plant sample. Hot water and steam are the main factors influencing the release of bioactive compounds from the plant tissue. In indirect water cooling, the mixture of water vapor and oil is condensed. The condensed mixture flows from the condenser to a separator, where the oil and bioactive compounds are automatically separated from water [57].

## 8. Unconventional Techniques

### 8.1. Ultrasound-Assisted Extraction (UAE)

Ultrasound-assisted extraction (UAE) uses ultrasound and solvents to extract compounds such as polyphenols, carotenoids, flavors, and polysaccharides from plant matrices (whole plants and by-products). Variables associated with VAE such as frequency, power, duty cycle, temperature, time, solvent type, and liquid–solid ratio must be precisely controlled for optimal extraction [58]. Ultrasound is a special type of sound wave. In chemistry, only a small part of the ultrasonic spectrum (between 20 kHz and 100 MHz) is used and referred to as ultrasonic power [59]. When the ultrasonic spectrum penetrates a medium, compression and expansion occur. This process leads to a phenomenon known as cavitation, i.e., the generation, growth, and collapse of bubbles. The collapsing bubbles generate shock waves and accelerated collision between particles, which leads to the fragmentation of the cell structure [60].

### 8.2. Pulsed Electric Field Extraction (PEF)

Treatment with pulsed electric fields (PEFs) improves diffusion and mass transport through a phenomenon known as the “electroporation” or “electropermeabilization” of membranes. This treatment externally supports the pressing, drying, extraction, and diffusion processes [61]. The principle of PEFs is to disrupt the cell membrane structure in order to increase extraction. Due to the dipolar nature of the membrane molecules, the electrical potential separates the molecules according to their charge in the cell membrane. Once a critical value of the transmembrane potential is exceeded, repulsion occurs between the charge-carrying molecules, forming pores in weak areas of the membrane and causing a drastic increase in permeability [62].

### 8.3. Enzyme-Assisted Extraction (EAE)

Enzyme-assisted extraction is a new and effective way to release bound compounds and increase the overall yield [63]. Some phytochemicals from plant matrices are dispersed in the cell cytoplasm and some compounds are retained in the polysaccharide–lignin network by hydrogen or hydrophobic bonds that are not accessible with a solvent in a routine extraction process [64]. In general, enzymes such as cellulase, α-amylase, and pectinase can be used as catalysts to support the extraction, modification, or synthesis of bioactive compounds in plants, as they can catalyze reactions with exquisite specificity and selectivity [65]. The enzyme-assisted extraction technique reduces the use of solvents and energy consumption and is said to be a more environmentally friendly and effective alternative process compared to conventional solvent extraction methods [66].

### 8.4. Microwave-Assisted Extraction (MAE)

Microwave-assisted extraction (MAE) uses non-ionizing microwaves with a frequency between 300 MHz and 300 GHz, which significantly intensify the extraction process. Microwaves can penetrate certain materials and interact with polar components to generate heat [67]. Electromagnetic energy is converted into heat via the mechanisms of ion conduction and dipole rotation and therefore only selective and specific materials can be heated based on their dielectric constant. In the ion conduction mechanism, heat is generated due to the resistance of the medium to the flow of ions. The ions maintain their direction along frequently changing field signals [68]. The mechanism of microwave-assisted extraction is believed to involve three sequential steps: first, the separation of solutes from the active sites of the sample matrix under higher temperature and pressure; second, the diffusion of the solvent through the sample matrix; and third, the release of solutes from the sample matrix into the solvent [69].

### 8.5. Pressurized Liquid Extraction (PLE)

Pressurized liquid extraction (PLE) has established itself as a green extraction technique with high throughput. Recently, PLE has become increasingly popular for the extraction of bioactive compounds (dietary fiber, various types of phenolic compounds and antioxidants, polyunsaturated fatty acids, amino acids, proteins, and minerals). This growing interest is mainly due to the fact that PLE is automated, extraction time and solvent consumption are reduced, and the configuration is tailored to oxygen- and light-sensitive analytes [70]. In general, the following five steps are performed: (1) wetting of the sample (analytes to be extracted and the matrix) with the extraction solvent; (2) desorption of the compounds from the matrix (including or without breaking chemical bonds); (3) solvation of the compounds in the extraction solvent; (4) dispersion of the compounds outside the matrix; and (5) diffusion through the nearest solvent layer around the matrix to finally reach the main solvent [71].

### 8.6. Supercritical Fluid Extraction (SFE)

The supercritical state of a solvent is reached when the temperature and pressure are above their critical points: a point above which no distinguishable gas and liquid phases exist [72]. As a result, it has gas-like diffusion, viscosity, and surface tension properties, while its density and solvation power are similar to liquids, making it suitable for the extraction of plant materials [73]. SFE is an ecological and highly selective method. Carbon dioxide (CO_2_) is the most common solvent for SFE, but other solvents such as ethylene, methane, nitrogen, xenon, or fluorocarbons are also used [74]. In SFE, the raw materials are kept in an extractor at a controlled temperature and pressure. The dissolved material is transferred to a separator. The extracts are then collected from the separator and the regenerated liquid is released into the environment. After extraction, the system is depressurized to convert CO_2_ from liquid to gas [75].

The formulation of biofungicides is an important process that must ensure minimal negative impact on undesirable organisms while maximizing the effect of the active ingredient [76]. Although biofungicides constitute an important sector of new products that contribute to agronomic safety, there are still challenges in formulations due to the degradation of the biomass or bioactive metabolite due to factors such as air, light, and temperature, as well as ensuring easy handling, application, and production feasibility [12]. Furthermore, from the laboratory stage, the development of a commercial biofungicide based on plant extracts consists of three complex phases: (i) the development of a viable and stable formulation; (ii) patent application; and (iii) registration of the active ingredient and its formulation [77], which considerably delays the release of the new product to the market.

## 9. Biofungicides in the Control of Phytopathogenic Fungi

The use of extract-based biofungicides in agricultural production systems brings important benefits to farmers, such as food security, reductions in phytopathogens, improvements in product quality, and higher prices [78]. It is important to mention that the plant-based chemical compounds contained in the extracts are not always specific to the target organism but are less toxic for them, especially for pollinating and predatory bees [79]. Depending on the plant source and the concentrations used, biofungicides have little or no allelopathic effects on crops [80]. Examples of some extracts with antifungal activity can be found in Table 2. The most important phytopathogenic fungi include the following:

*Monilinia fructicola* is a pathogen responsible for losses in crops such as peach (*Prunus pérsica* L.), apricot (*P. armeniaca* L.), plum (*P. domestica* L.), almond (*P. amygdalus* Batsch), apple (*Malus pumila* Mill.), and pear (*Pyrus communis* L.) [111]. The pathogen infects the aerial parts of host plants with a variety of symptoms, including the wilting of flowers, cankers in woody tissues, and rotting of fruit [112]. In the field, the incidence of brown rot increases as harvest time approaches, and the fruits are also more susceptible to infection. In general, the fruit that arrives at the packing houses appears to be disinfected, but in reality, it could be contaminated by conidia on its surface or by conidia that have already infected fruit in the orchard but without visible symptoms [113]. Hernandez et al. [91] used a purified polyphenolic extract of orange peel (*Citrus × sinensis* L.) at different concentrations for 8 days to stop the growth of *M. fructicola* and reported fungicidal activity attributable to ferulic acid (Table 2). In another study, Pazolini et al. [98] obtained aqueous extracts of rapeseed (*Brassica napus* L.) and Indian mustard (*Brassica juncea* L.) and also investigated their biological activity on *M. fructicola*. The growth of the mycelium of *M. fructicola* and the germination of the conidia were reduced on average by 95 and 31%, respectively. The antifungal activity was attributed to glucosinolates, such as nitriles, thiocyanates, and epinitriles, but mainly to isothiocyanates (Table 2).

*Fusarium oxysporum*: It is well represented in soil fungal communities; all strains of *F. oxysporum* are saprophytic and can grow and survive for long periods in soil organic matter and in the rhizospheres of many plant species. In general, this phytopathogen penetrates the roots and causes rot in the vascular system of crops and ornamental plants [114]. Ramírez et al. [92] obtained extracts from the leaves of *Stevia rebaudiana* Bertoni and seven fractions to test their antifungal activity in vitro and in tomato plants (*Solanum lycopersicum* L.) inoculated with *F. oxysporum*. In this study, an in vitro growth inhibition of more than 50% was observed, while in vivo tests showed that all tomato plants treated with the extract were larger and had a higher dry weight in air and roots than the other plants. Of the treatments, this activity was attributed to austroinulin (Table 2). Hernández Soto et al. [97] tested the activity of *Argemone ochroleuca* L. leaves against *F. oxysporum*, finding a growth inhibition of over 60%. This activity was attributed to compounds such as berberine, isoquinoline, ehydrocorydalmin, and oxyberine.

*Colletotrichum* spp. include several plant pathogens of major importance that cause disease in a wide range of woody and herbaceous species, although there are some known species that attack crops of commercial value, such as strawberry (*Fragaria vesca* ex Weston), mango (*Mangifera indica* L.), avocado (*Persea Americana* Mill), corn (*Zea mays* L.), sugarcane (*Saccharum officinarum* L.), and sorghum (*Sorghum* L.). This genus is the eighth most important group of phytopathogenic fungi in the world [115]. As plant pathogens, *Colletotrichum* species are primarily described as causing anthracnose diseases, although other diseases such as red rot of sugarcane (*Saccharum officinarum* L.), coffee berry disease (*Coffea arabica* L.), crown rot of strawberry (*Fragaria vesca* ex Weston) and banana (*Musa × paradisiaca* L.), and brown spot disease of bean (*Phaseolus vulgaris* L.) [116]. Disease symptoms of anthracnose include limited, often sunken, necrotic lesions on leaves, stems, flowers and fruits; crown and stem rot; and seedling blight [117]. Gasca et al. [93] investigated the in vitro activity of the fruit extract of *Sapindus saponaria* L. against three species of *Colletotrichum*: *C. musae*, *C. gloeosporioides*, and *C. boninense*. The activity of the extract against *C. musae* was similar to that of thiabendazole (conventional fungicide). This activity was attributed to compounds such as the saponin 3-O-(β-d-xylopyranosyl)-(1→3)-α-l-rhamnopyranosyl-(1→2)-α-l-arabinopyranosyl hederagenin and acyclic sesquiterpenoligoglucoside (Table 2). Dudoit et al. [94] investigated the in vitro and in vivo antifungal activity of an extract of Brazilian red propolis (*Sesuvium portulacastrum* L.) against *Colletotrichum musae*. The extract of Brazilian red propolis inhibited 42% of the growth of *C. musae* in in vitro tests. When compounds were identified, medicarpin, (3S)-vestitol, and (3S)-neovestitol were reported to be the compounds with the highest antifungal activity. Finally, the in vivo results showed that the extract could be a very interesting candidate for an alternative treatment to chemical fungicides in the control of banana crown rot. De Rodríguez et al. [101] investigated the inhibitory activity of extracts from *Lippia graveolens* Kunth, *Agave lechuguilla* Torr., *Yucca carnerosana* (Trel.) Mc Kelvey, and *Yucca filifera* Chaub against *Rhizopus stolonifer*, *Colletotricum gloesporoides*, and *Penicillium digitatum* in vivo. The results showed that the extracts of *L. graveolens* exhibited 100% fungicidal activity against *R. stolonifer.*

*Rhizoctonia solani* represents an economically important group of soilborne basidiomycete pathogens that occur on many plant species worldwide [118]. *R. solani* causes root rot and leaf blight. Heavily infested crops include potato (*Solanum tuberosum* L.), soybean (*Glycine max* L.), bean (*Phaseolus vulgaris* L.), pea (*Pisum sativum* L.), tomato (*Solanum lycopersicum* L.) and watermelon (*Citrullus lanatus* (Thunb.) Matsum & Nakai var. Lanatus) [119]. Al-Rahmah et al. [100] evaluated the fungicidal activity of five methanol extracts of the following plants: *Lantana camara* L., *Salvadora persica* L., *Thymus vulgaris* L., *Zingiber officinale* Roscoe, and *Ziziphus spina-christi* L. on the phytopathogenic fungi *Fusarium oxysporum*, *Pythium aphanidermatum*, and *Rhizoctonia solani*. The extracts of *T. vulgaris* and *Z. officinale* were highly active and showed fungistatic and fungicidal activity against phytopathogenic fungi. The biological activity was attributed to compounds such as thymol, carvacrol, β-cymene, α-terpinolene, gingerol, cedrene, zingiberene, and α-curcumene (Table 2). Hernández et al. [104] investigated the antifungal activity of the essential oils of *Lantana achyranthifolia* L. and *Lippia graveolen* L. against *Fusarium monoliforme* and *Rhizoctonia solani*; *Lippia graveolens* showed greater antifungal activity. This was attributed to compounds such as carvacrol, α-terpinyl acetate, m-cymene, and thymol (Table 2). De Rodríguez et al. [106] investigated the antifungal activity of aqueous and ethanolic extracts of *Flourensia microphylla* (A.Grey), *Flourensia cernua* S.F. Blake, and *Flourensia retinophylla* S.F. Blake against *Alternaria* sp., *Rhizoctonia solani*, and *Fusarium oxysporum*. The three species exhibited fungicidal activity. The chemical composition indicates that the activity is due to compounds such as Benzofurans, Benzopyrans, Dehydrofluorenic acid, Flourensadiol, and Methylorselinate (Table 2).

*Alternaria alternata* causes black spots on many fruits and vegetables around the world. It is a latent fungus that develops during cold storage of fruits and becomes visible during the marketing period, causing large postharvest losses [120]. *A. alternata* is widespread in many regions of the world. This fungus has been found to be responsible for various diseases during the postharvest period of many horticultural crops. Reported diseases include stem-end rot of mangoes (*Mangifera indica* L.), black rot of cherry tomatoes (*Solanum lycopersicum* var. Cerasiforme), golden cores and moldy cores of apples (*Malus domestica* L.), fruit spots on apples, black rot on mandarins (*Citrus reticulata*, Blanco), black rot on kiwifruit (*Actinidia delicious* (A.Chev.) C.F.Liang and A.R.Ferguson), and black spots on melons (*Cucumis melo* L.), among others [121]. Hernández et al. [91] investigated the antifungal capacity of the polyphenolic extract of orange peel against: *Monilinia fructicola, Botrytis cinerea* and *Alternaria alternata*. The extract inhibited (100%) the mycelial growth and germination of the conidia of the three fungi. This antifungal activity is mainly due to the phenolic compounds present in the orange peel: flavonoids (naringin, hesperidin, and neohesperidin), phenolic acids (ferulic acid and p-coumaric acid), ferulic acid, and p-coumaric acid (Table 2). Despite the various reports of extracts with antifungal activity, there is still no consensus on the evaluation concentrations that allow classifying the biological response into active and inactive, which means that there is no defined range of concentrations among authors to classify a plant extract as a promising antifungal. According to what was reported by Mesa et al. [122], it is recommended to establish a classification consensus according to the effect of the extract as either active, moderately active, slightly active, or harmless (Figure 4).

## 10. Regulation, Advantages, Disadvantages, and Current Panorama of Biofungicides in Agriculture

Although scientifically sound tests have been carried out on the fungicidal properties of plant products and their components, and some disease control products based on plant extracts or essential oils have also been launched [123], it does not seem sufficient to replace them with the current synthetic fungicides, as they do not always provide the desired disease control in the field [124]. Biofungicides can be used in combination with synthetic fungicides to provide disease control and gradually change the use of conventional fungicides. The introduction and widespread use of biofungicides enables the production of food with no or minimal fungicide residues [125].

Biofungicide residues are less harmful to living organisms and the environment and are relatively safe, even when applied just before harvest. This helps growers to meet consumer demand for more natural, healthier, and safer food in relation to the use of synthetic fungicides [126]. However, it is important to mention that biofungicides are generally expensive compared to synthetic fungicides [127]. The commercialization of biofungicides is significantly disadvantaged by the registration systems [128]. Specific criteria required for the commercial use of these products include toxicity, production efficiency, and product safety [129]. Obtaining this information from companies can be very costly and may discourage them from commercializing biofungicides [130]. Part of the challenge with registration is that the guidelines for biopesticide registration rely too heavily on the criteria used for chemical fungicides and require information that is not as readily available for biofungicides. Toxicological and environmental risk assessments are also foreseen, but these are very expensive, especially when the size of the market in certain countries does not justify the cost [131]. In addition to the lengthy registration process, the cost of commercialization and time to market also discourage small- and medium-sized companies.

In Europe, biopesticides are regulated according to the same standards that apply to chemicals. Harmonization of the requirements and interpretation of registration data for biopesticides occurred with the development of Council Directive 91/414/EEC concerning the placing of plant protection products on the market and the Biocidal Products Directive 98/8/EC, which covers the requirements for oils and extracts [132,133]. Since the adoption of Council Directive 91/414/EEC, which was later replaced by Regulation (EC) No 1107/2009, which forms the backbone of the authorization procedure for biofungicides, it has been possible to register biorational substances as “low-risk” or “basic” according to the procedure described in Articles 22 and 23 of this Commission document, which provides two registration routes for SMEs (small- and medium-sized enterprises) [134]. In South Africa, the Department of Agriculture regulates the use of biofungicides (DALRRD) through the Land Reform and Rural Development—Guidelines for the Registration of Biological Agricultural Products in South Africa 2015 [135], in Kenya, the Pest Control Products Board (PCPB) regulates use through the Pest Control Products Act [Act No. 4 of 1982, L.N. 89/1983, Act No. 6 of 2009] [136], and in Nigeria, the National Agency for Food and Drug Administration and Control (NAFDAC) regulates use through the Biopesticide Registration Regulations 2019 [137].

In North America, the Organic Materials Review Institute (OMRI) plays a crucial role in this process. The OMRI is an organization dedicated to the review and certification of products used in organic production (including disease control products) and processing to ensure that they meet organic standards. OMRI standards are based on the standards of the US National Organic Program (NOP), the standards of the Canadian Organic Regime (COR), and the guidelines of the Mexican Organic Products Act (LPO). The testing and approval process usually takes several months. If a product meets the criteria, it is OMRI-listed^®^ and added to the OMRI© product lists, giving organic farmers and consumers the assurance that it is truly organic and safe for human consumption and the environment. Atmosphere. All products must be renewed annually [138].

For a biofungicide to be approved, it is essential to assess its safety and efficacy. However, many investors see regulation as an obstacle to the development and commercialization of the product. Field trials and risk assessments, which usually take two growing seasons, are a major challenge. In addition, the different modes of action of some microorganisms require a large amount of data to establish safety and efficacy criteria. Many applicants do not complete the registration of plant protection products (PIPs) as biopesticides, probably due to difficulties in assessing the safety of GMOs. Successful commercialization also faces challenges as biopesticides must compete in terms of cost, acceptance, and efficacy with conventional pesticides, which are better known, less expensive, and faster-acting. Political and social constraints further complicate commercialization [139].

Only larger companies are able to bear the research costs required to prepare a registration document [140]. The market for agrochemicals clearly shows the impact of current regulations on consumer use and preferences. Although sales of chemical fungicides declined by 1.3% annually, the growth rate for the biofungicide market is 15.6% [141]. Organic farming is practiced in 188 countries and more than 96 million hectares of agricultural land are farmed organically by at least 4.5 million farmers. Global sales of organic food reached almost EUR 135 billion in 2022 [138]. Currently, organic farming continues to grow, highlighting the importance of creating environmentally friendly alternatives for crop production [142].

## 11. Conclusions and Future Prospects

Due to the impact that synthetic fungicides have on the environment and health, the use of these products will need to be strictly regulated by governments in the near future, which may lead to an increase in demand for plant-based products. This makes it clear that plant extracts are effective, biodegradable, and not as harmful to the environment as the synthetic chemicals that are often used. The option of replacing chemicals with plant-based formulations fits in well with a future-oriented food and agricultural policy. Agriculture cannot rely on the use of synthetic fungicides, as has long been the case in developing countries. Local resources must be used. Therefore, the production of more biofungicides must become common practice and the approval policy for their commercialization must also be regulated.

## Figures and Tables

**Figure 1 ijms-25-06879-f001:**
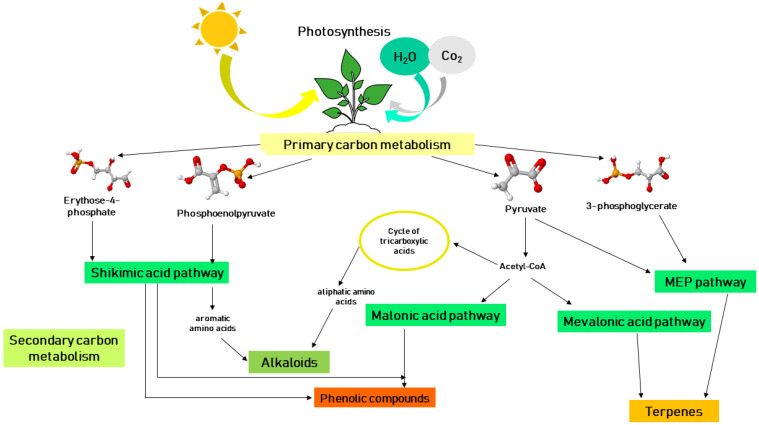
Metabolic pathways by which secondary metabolites are synthesized in plants.

**Figure 2 ijms-25-06879-f002:**
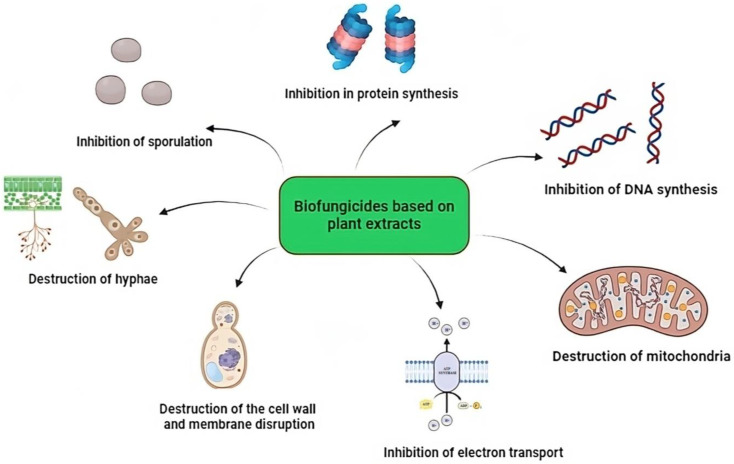
Mechanisms of action of plant-based biofungicides against phytopathogenic fungi.

**Figure 3 ijms-25-06879-f003:**
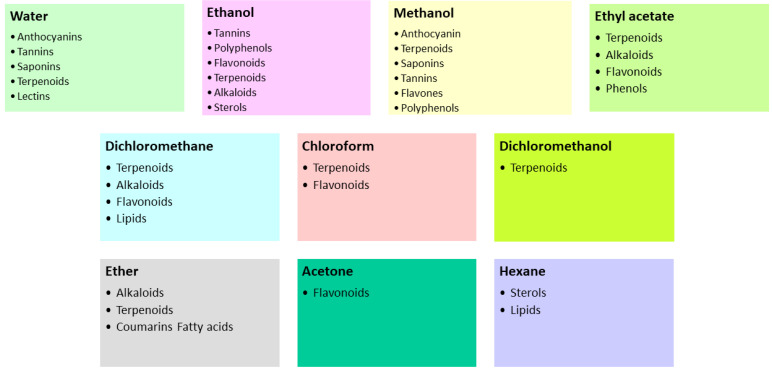
Extracted bioactive compounds by different solvents.

**Figure 4 ijms-25-06879-f004:**
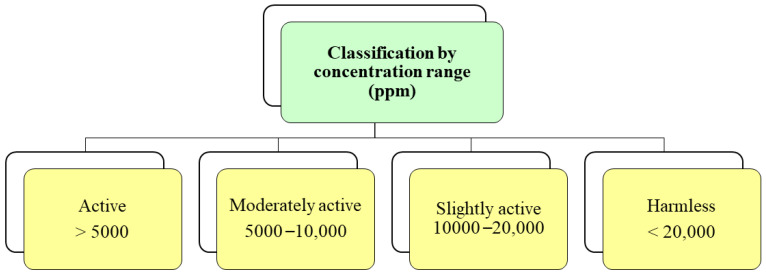
Classification according to the effect of the plant extract on phytopathogenic fungi.

**Table 1 ijms-25-06879-t001:** Secondary metabolite extraction techniques.

Techniques	
Conventional Extraction	Non-Conventional Extraction
Soxhlet extraction	Ultrasound-assisted extraction (UAE)
Maceration	Pulsed electric field extraction (PEF)
Hydrodistillation	Enzyme-assisted extraction (EAE)
	Microwave-assisted extraction (MAE)
	Pressurized liquid extraction (PLE)
	Supercritical fluid extraction (SFE)

**Table 2 ijms-25-06879-t002:** Plant extracts for the control of phytopathogenic fungi.

Phytopathogenic Fungus	Disease	Vegetable Extract	Bioactive Compound with Biological Activity	Reference
*Colletotrichum acutatum*	Anthracnose	Ethanolic Garlic peel extract	Allyl trisulfide; allyl methyl trisulfide; allyl disulfide; allyl trans-1-propenyl disulfide; allyl methyl sulfide; and 2-vinyl-4H-1,3-dithiine	[81]
*Alternaria alternata*	Leaf-spot	Aqueous *Dittrichia viscosa* L. extract	Caffeoyl quinic acids; methoxylated flavonoids; sesquiterpene	[82]
*Botrytis cinerea*; *Colletotrichum acutatum*; *Diplodia corticola*; *Phytophthora cinnamomic*; *Fusarium culmorum*	Gray mold; anthracnose; trunk canker; chestnut ink; vascular wilting	Hydroethanolic extract *Myristica fragans* Houtt	Tetradecanoic acid, 9-octadecenoic acid; *n*-hexadecanoic acid; dodecanoic acid; octadecanoic acid; veratone; gelsevirine; and montanine	[83]
*Botrytis cinerea*; *Colletotrichum acutatum*; *Diplodia corticola*; *Phytophthora cinnamomic*; *Fusarium culmorum*	Gray mold; anthracnose; trunk canker; chestnut ink; vascular wilting	Hydroethanolic *Curcuma longa* L. extract	*β*-turmerone; *α*-turmerone; -*ar*-turmerone; *α*-atlantone; *γ*-curcumene; zingiberene; isoelemicin; and gibberellin	[84]
*Alternaria alternata*; *Fusarium solani*; *Fusarium oxysporum*	Black mold; vascular wilting	Methanolic *Cinnamomum camphora* (L.) J. Presl extract	Mono(2-ethylhexyl) ester of 1,2-benzene dicarboxylic acid	[85]
*Alternaria alternata*; *Fusarium oxysporum*; *Rhizoctonia solani*	Black mold; vascular wilting; damping-off	Polyphenol extracts: Mesquite (*Prosopis glandulosa* Torr), Cedar (*Cedrus* Trew), and Oak (*Quercus* L.)	Anthocyanins; flavonols and flavones	[86]
*Fusarium oxysporum*; *Rhizoctonia solani*	Vascular wilting; damping-off	Methanolic *Eryngium campestre* L. extract	Benzoic acid; catechol; quercetin; vanillic acid; resveratrol; naringenin; and quinol	[87]
*Fusarium oxysporum*, *Fusarium solani* *Phytophthora capsici*	Vascular wilting; pepper blight	Essential oil: Cinnamon (*Cinnamomum zeylanicum* J. Presl), Neem (*Azadirachta indica* A.Juss ) oil, Sapote (*Diospyros digyna* (J.F.Gmel.) Perr) extract	Cinnamaldehyde; kaempferol; cinnamic alcohol; alkaloids; essential oils; polyphenols; tannins; and saponins	[88]
*Pestalotiopsis clavispora*; *Colletotrichum gloeosporioides*; *Lasiodiplodia pseudotheobromae*	Crown rot; antracnosis; gummosis	Mixture of ethanol and ethyl acetate: *Lantana hirta* L., *Argemone ochroleuca* L. and *Adenophyllum porophyllum* (Cav.) extract	L. *hirta* extract: phytol and α-Sitosterol. In *A. ochroleuca*: toluene and benzene, 1,3-bis(3-phenoxyphenoxy)-. In *A. porophyllum*: hexanedioic acid, bis(2-ethylhexyl) ester	[89]
*Ralstonia solanacearu*; *Phytophthora infestans*; *Neopestalotiopsis javaensis*	Banana bacterial wilt; late blight; scab diseases	Methanol *Pernettya prostrata* (Cav.) and *Rubus roseus* Schott	Phenolic compounds	[90]
*Monilinia fructicola*; *Botrytis cinerea*; *Alternaria alternata*	Brown rot; gray mold; black mold	Orange (*Citrus × sinensis* L.) peel polyphenolic extract	Ferulic acid and p-coumaric acid	[91]
*Fusarium oxysporum*	Vascular wilting	Hexane extract of *Stevia rebaudiana* Bertoni leaves	Austroinulin	[92]
*Colletotrichum* spp.	Anthracnose	Ethanolic extract of the fruit of *Sapindus saponaria* L.	Saponin 3-O-(β-d-xylopyranosyl)-(1→3)-α-l-rhamnopyranosyl- (1→2)-α-l-;arabinopyranosyl hederagenin and acyclic sesquiterpene oligoglycoside	[93]
*Colletotrichum musae*	Anthracnose	Propolis ethanolic extract of Brazilian Red (*Sesuvium portulacastrum* L.)	Medicarpin, (3S)-vestitol, and (3S)-neovestitol	[94]
*Monilinia laxa*; *M. fructigena*	Brown rot	Pomegranate (*Punica granatum* L.) peel aqueous extract	Phenolic and flavonoid compounds	[95]
*Botryosphaeria dothidea*; *Colletotrichum musae*; *Pestalotipsis guepinii*; *Colletotrichum orbiculare*; *Phylophthora nicotianae*; *Pestalotiopsis longiseta*; *Sclerotinia sclerotiorum*	Canker; anthracnose; leaf spots; anthracnose; late blight; peduncular rots; white mold	The dichloromethane extract of *Waltheria* *Indica* L.	Alkaloids; antidesmone; and waltherione C	[96]
*Colletotrichum gloesoporioides*, *Fusarium oxysporum*, *Rhizoctonia solani*	Anthracnose; vascular wilting; damping-off	Methanol extract of leaves of *Argemone ochroleuca* L.	Berberine; Isoquinoline; Ehydrocorydalmine; and Oxyberberine	[97]
*Monilinia fructicola*	Brown rot	Aqueous extracts of *Brassica napus* L. and *Brassica juncea* L.	Glucosinolates: nitriles, thiocyanates, epinitriles; but, mainly, isothiocyanates	[98]
*Fusarium oxysporum*	Vascular wilting	Ethanolic extract of *Rhus muelleri* Müller	Ethyl iso-allocholate (steroid); 7,8-epoxylanostan-11-ol,3-actoxy (alcoholic compound); and 3-trifluoro acetoxy pentadecane (flourcompound)	[99]
*Fusarium oxysporum*; *Pythium aphanidermatum*; *Rhizoctonia solani*	Vascular wilting; mildew; damping-off	Methanolic extracts of the leaves of *Thymus vulgaris* L. and *Zingiber officinale* (Rosc., shengjiang)	Timol; carvacrol; β cimeno; α-terpinoleno; gingerol; cedreno; zingibereno; and αcurcumeno	[100]
*Rhizopus stolonifer*; *Colletotricum gloesporoides*; *Penicillium digitatum*	Soft rot; anthracnose; green rot	Ethanolic and hexane extracts of *Lippia* graveolens Kunth, *Agave lechuguilla* Torr, *Yucca carnerosana* (Trel.) Mc Kelvey, and *Yucca filifera* Chaub	Saponins, tannins, and flavonoids	[101]
*Colletotrichum acutatum*; *C. fragariae; C. gloeosporioides*	Anthracnose	Essential oil of *Arnica longifolia* L., *Aster hesparius* Michx, and *Chrysothamnus nauseous* L.	R-bisabolol and carvacrol	[102]
*Fusarium oxysporum*; *Alternaria alternata*; *Geotrichum candidum*; *Penicillum digitatum*; *Aspergillus niger*	Vascular wilting; black mold; rot; green mold; black mold	Aqueous extracts of *Lippia berlandieri* L.	Eugenol	[103]
*Fusarium monoliforme*; *Rhizoctonia solani*	Panicle blight; damping-off	Essential oil of *Lantana achyranthifolia* L. and *Lippia graveolens* L.	Carvacrol; α-bisabolol, isocaryophyllene, and thymol	[104]
*Fusarium oxysporum*	Vascular wilting	Essential oil of *Chenopodium ambrosioides* L.	α-terpinene; P-cimene; 4-carene; and Trans-ascaridol	[105]
*Alternaria* sp.; *Rhizoctonia solani*; *Fusarium oxysporum*	Leaf spots; damping-off; vascular wilting	Ethanol extracts of *Flourensia microphylla* (A.Gray), *Flourensia cernua* S.F. Blake, *Flourensia retinophylla* S.F. Blake	Benzofurans, benzopyrans, dehydrofluorenic acid, flourensadiol, methyl orselinate	[106]
*Fusarium graminearum*; *Fusarium solani*; *Fusarium verticillioides*; *Macrophomina phaseolina*	Head blight; stem base rot in vegetables; ear rot; collar rot in soybeans	Chloroform Extract of *Larrea divaricata*	Apigenine-7-methylether; nordihydroguaiaretic acid; 3,4’-dihydroxy-3’,4-dimethoxy-6,7’-cyclolignan	[107]
*Alternaria alternata*, *Fusarium oxysporum*, *Phoma destructive*, *Rhizoctonia solani Sclerotium rolfsii*	Black mold; vascular wilting; leaf spot; damping-off; root and stem rot	The ethanolic extract of seeds of *Alhagi maurorum* Medik	Flavonoids; glycosides; alkaloids; saponins; tannins; steroids; and anthraquinone	[108]
*Phytium ultimum*	Root rot	Essential oil of *Ocotea quixos Lauraceae* Lam	trans-Cinnamaldehyde	[109]
*Alternaria alternata*; *Botrytis cinerea*; *Fusarium oxysporum*; *Penicillium expansum*	Black mold; gray mold; vascular wilting; blue mold	Five different extracts of each seaweed (*n*-hexane, ethyl acetate, aqueous, ethanolic, and hydroethanolic)	Polysaccharides commonly present in seaweeds, such as laminarin fucoidans or alginates and phlorotannins	[110]

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
