# Peer review of "Biofungicides Based on Plant Extracts: On the Road to Organic Farming"

_ijms, 2024, doi:10.3390/ijms25136879_

Round 1

Reviewer 1 Report

Comments and Suggestions for Authors

Biofungicides are effective against plant diseases and are more environmentally friendly than synthetic fungicides. The use of biofungicides is also increasing around the world . This article  outline biofungicides based on plant extracts biosynthesis from secondary metabolites, extraction techniques and formulation of biofungicides, the biological activity of plant extracts on phytopathogenic fungi, the mechanism of action and the current panorama of biofungicides in agriculture. It is helpful for people to understand the mechanism of biofungicides, its importance and the difficulties in promoting its use at this stage. However, in order to improve the quality of the article, the following suggestions are given:

1. Chemical pesticides have better efficacy than biological pesticides and are more conducive to increasing the yield of agricultural products in order to supply the growing population. The introduction should focus on the safety of biofungicides, and add more content to supplement the benefits of biological pesticides on environmental protection and human health promotion.

2. In line68, “ in agriculture. Organic.” changed to “ in organic agriculture.”

3. "3. Terpenes", "4. Alkaloids" and "5. Phenolic compounds" are part of "2. Biosynthesis of secondary metabolites in plants", suggesting merging in one part. This paper introduced the mechanism of action and synthesis of terpenes, alkaloids and phenolic compounds, but does not introduce the related fungicide products, it is recommended to supplement.

4. In addition to the solvents mentioned in Table1, there are also dichloromethane, ethyl acetate and other solvents for extracting bioactive substances. It is recommended to search relevant literature and supplement the solvent. At the same time, the extraction method of biofungicides is suggested to be added to the paper.

5. “The plant-based chemical compounds contained in the extracts are specific to the target organism” is not correct. The plant-based chemical compounds usually are not necessarily targeted.

6. “Biofungicides are usually available at relatively low prices compared to the commercial fungicides currently in use” is not correct. Biofungicides are usually expensive compared to the synthetic fungicides.

7. There is no emphasis on "on the road to organic farming". It is suggested to describe the difficulties and advantages of biofungicides in moving towards organic agriculture, compare the relevant regulations in different countries, and give substantive suggestions.

Comments on the Quality of English Language

The manuscript is well written.

Reviewer 2 Report

Comments and Suggestions for Authors

I. Hernandez-Soto and collaborators propose a review, with 97 references, dealing with the potential of bioresources, more precisely secondary metabolites from plants, to have fungicide activities. This review is organized in 8 chapters and focuses on 4 main molecular families: terpenes, alkaloids, phenolic compounds. A chapter is also dedicated to the main principles of extraction procedures of secondary metabolites and the final one to the control of phytopathogenic fungi.

This review is of interest and well written. Some descriptions of the inhibition pathways would have deserved to be described in more depth.

Minor corrections:

-          L36-38: a reference is required

-          L68: remove organic

-          L91: methyl-d- instead of methyl- d –

-          L109: Rephrase this sentence with a verb

-          Fig2: Non-conventional instead of no- conventional

-          L243: replace mcymene by m-cymene

-          L244: which extract of Flourensia? Precise solvent, fraction, etc.

-          L303: precise the stereochemistry of the sulphoxide function

Round 2

Reviewer 1 Report

Comments and Suggestions for Authors

The author has answered all my questions and revised the article according to my suggestions, and I think the article is acceptable.

But I still think "3. Terpenes", "4. Alkaloids" and "5. Phenolic compounds" are part of "2. Biosynthesis of secondary metabolites in plants" and suggest merging in one part.